# Association Analysis of Salt Tolerance in Asiatic cotton (*Gossypium arboretum*) with SNP Markers

**DOI:** 10.3390/ijms20092168

**Published:** 2019-05-01

**Authors:** Tussipkan Dilnur, Zhen Peng, Zhaoe Pan, Koffi Kibalou Palanga, Yinhua Jia, Wenfang Gong, Xiongming Du

**Affiliations:** State Key Laboratory of Cotton Biology, Institute of Cotton Research, Chinese Academy of Agricultural Sciences, Anyang 455000, China; tdilnur@mail.ru (T.D.); cripengzhen09@126.com (Z.P.); panzhaoe@163.com (Z.P.); palangaeddieh@yahoo.fr (K.K.P.); jiayinhua_0@sina.com (Y.J.); gwf018@126.com (W.G.)

**Keywords:** *Gossypium arboretum*, salt tolerance, single nucleotide polymorphisms, association mapping.

## Abstract

Salinity is not only a major environmental factor which limits plant growth and productivity, but it has also become a worldwide problem. However, little is known about the genetic basis underlying salt tolerance in cotton. This study was carried out to identify marker-trait association signals of seven salt-tolerance-related traits and one salt tolerance index using association analysis for 215 accessions of Asiatic cotton. According to a comprehensive index of salt tolerance (CIST), 215 accessions were mainly categorized into four groups, and 11 accessions with high salinity tolerance were selected for breeding. Genome-wide association studies (GWAS) revealed nine SNP rich regions significantly associated with relative fresh weight (RFW), relative stem length (RSL), relative water content (RWC) and CIST. The nine SNP rich regions analysis revealed 143 polymorphisms that distributed 40 candidate genes and significantly associated with salt tolerance. Notably, two SNP rich regions on chromosome 7 were found to be significantly associated with two salinity related traits, RFW and RSL, by the threshold of −log_10_*P* ≥ 6.0, and two candidate genes (Cotton_A_37775 and Cotton_A_35901) related to two key SNPs (Ca7_33607751 and Ca7_77004962) were possibly associated with salt tolerance in *G. arboreum*. These can provide fundamental information which will be useful for future molecular breeding of cotton, in order to release novel salt tolerant cultivars.

## 1. Introduction

Soil salinity accumulation has become a serious environmental problem [1] that could negatively affect plant growth, geographical distribution, and agricultural products [2,3]. Salinization consists of the accumulation of water-soluble salts in the soil, including ions of potassium (K^+^), magnesium (Mg^2+^), calcium (Ca^2+^), chloride (Cl^−^), sulfate (SO_4_^2−^), carbonate (CO_3_^2−^), bicarbonate (HCO_3_^−^) and sodium (Na^+^). The causes of land salinization can be divided into two categories: 1) primary (natural) and 2) secondary (anthropogenic) [4]. The primary reason includes arid climates, high underground water levels, seawater infiltration, and so on [5]. The secondary reason is irrigation practices. Soil salinization is reducing the area that can be used for agriculture by 1%–2% every year, hitting hardest in the arid and semi-arid regions. Therefore, the development of salt-tolerant crops is a pressing scientific goal [6], but the ability of plants to deal with these adverse factors is different [7]. Cotton is one of the advantageous salt-tolerant crop with a threshold salinity level of 7.7 dS·m^−1^ [8]. However, high salt concentrations can still hinder growth during the germination and seedling stages, which are the two most susceptible stages of plants [2,9].

The mechanisms involved in the response to salinity in cotton have been well described by Peng et al. [3,10]. Under salinity stress conditions, soluble salts are accumulated in the root zone of plants, then causing osmotic and ionic stress and mineral perturbations [3,11], leading to dramatic reductions in crop quality and yield [5]. However, the genetic control of salt tolerance is only partially understood, because of the diversity of the regulation mechanisms, and the complexity of the genetic architecture of salt tolerance [9]. As the fundamental aim of genetics is to connect genotype to phenotype, the identification and characterization of genes associated with agronomical important traits is essential for both understanding the genetic basis of phenotypic variation and efficient crop improvement. Modern molecular biology techniques and new statistical methods have opened new horizons for the cotton breeders; thus, linkage mapping and association mapping are the two important methods employed for QTL analysis. Molecular marker-quantitative trait association is one of the powerful approaches for exploring the molecular basis of phenotypic variations in plant [12], and could be used to increase the efficiency of a breeding program, especially for salinity tolerance [13,14]. The present studies of genetic map construction are mainly reflected in three different DNA based molecular markers such as simple sequence repeats (SSR) [15], single-nucleotide polymorphism (SNP) markers [16,17] and Intron length polymorphisms (ILD) markers [18].

Single Nucleotide Polymorphism is often abbreviated to SNP; it describes a variation in a single nucleotidae that occurs at a specific position in the genome [19]. Genome-wide association studies based on linkage disequilibrium (LD) is an effective strategy tool to study phenotype-genotype association. Compared with traditional QTL mapping, GWAS can use SNPs obtained by genome re-sequencing as molecular markers to dissect complex traits [20]. One SNP occurs every 100–300 bp in any genome; therefore SNPs markers have higher polymorphism than SSRs and other molecular markers [19]. GWAS has been successfully applied in rice, Arabidopsis, maize, wheat, barley and other crops to identify characteristic-related SNP markers of their important trait [21,22,23,24]. In cotton, various genetic maps based on SSR and SNP markers have been constructed using bi-parental mapping populations and natural population of *Gossypium hirsutum*; however, there are fewer studies, and no causal genes responsible for the salt tolerance traits from *Gossypium arboretum* have been identified [25,26,27,28].

Asiatic cotton (*Gossypium arboretum*) was introduced into China from ancient India, Burma or Vietnam over 2000 years ago [29]. Du et al (2018) reported that the natural population of Gossypium. arboretum was classified into three main groups represented South China, Yangtze River region, and Yellow River region groups respectively that exhibited strong geographical distribution [30]. A draft genome of cotton diploid *Gossypium arboretum* (the size is 1.7 Gb,2*n* = 2× = 26) was recently reported by Li et al. (2015) [31]. The genetic basis of Asiatic cotton will provide a fundamental resource for genetics research of the important agronomic traits for cotton breeding. Therefore, the present study was performed in consideration of the following objectives: (i) to screen salinity tolerance at germination stage; (ii) to analyze the marker-trait associations by using SNP markers; (iii) to identify the causal genes which are responsible for the salt tolerance traits from *G. arboreum*. 

## 2. Results

### 2.1. Phenotypic Diversity of G. arboretum Population

Seven salt tolerance related traits, including GR, FW, SL, WC, ChlC, EC, and MDA, were measured for all 215 *G. arboretum* accessions under 0 mM (C) and 150 mM (S) NaCl treatment (Figure 1 and Appendix A). ANOVA analysis of seven salt-tolerance-related traits as measured for genetic diversity shows significant difference among the accessions (*P* < 0.0001) (Table 1). Correlation of GR with FW and SL was highly significant (*P* < 0.001), while GR with ChlC was also significant (*P* < 0.01). Correlation results of FW with SL, ChlC and MDA were also highly significant with *P* < 0.001. Correlation between SL and ChlC was highly significant (*P* < 0.001). A positive correlation was found between ChlC and EC (*P* < 0.01), while a negative correlation was found between ChlC and MDA (*P* < 0.05). The correlation between related EC and MDA was significant at *P* < 0.05. Interestingly, correlation analysis revealed that WC had significant correlation (*P* < 0.05) with MDA (Table 2). According to the CIST, 215 accessions were mainly categorized into four clusters. Cluster 1 contained 12 accessions that were highly sensitive to salt treatment (<0.6), Cluster 2 contained 26 accessions that were moderate tolerant to salt treatment (0.6–1.5), Cluster 3 included 153 accessions that were tolerant (1.5–2.5), and Cluster 4 had 24 accessions that were highly tolerant to salt treatment (>2.5) (Figure 1, Appendix A). Based on this result, the high tolerant accessions (24) were selected (Appendix A). Basing on the comprehensive index of salt tolerance, we finally selected 11 high tolerant accessions (top 5%) for breeding using, including GuangXiZuoXianZhongMian, LiaoYang-1, ZhaoXianHongJieMian, PingLeXiaoHua, KaiYuanTuMian, YuXi33, ChangShuXiaoBaiZi, PingGuoJiuPingZhongMian, FuChuanJiangTangZhongMian, TangShanBaiZiZhongMian, and ShiJiaZhuangJianMian (Appendix A).

### 2.2. Association Mapping of Salt Tolerance Related Traits Using SNP Markers

The total of 1,568,133 high-quality SNPs (MAF > 0.05, missing rate < 40%) in 215 *G. arboreum* accessions were used for GWAS of the salt traits. The SNP markers associated with the seven salt-tolerance-related traits and one salt tolerance index were identified based on the threshold value, log_10_(*P*) ≥ 4.0, using the MLMM model in the EMMAX software (Appendix A). The threshold of −log_10_*P* ≥ 4.0 was also derived from the quantile–quantile (QQ) plots, For RGR, −log_10_(*P)* values and QQ plots suggested relatively weak genetic association (Figure 2a). Most of the upward deviation from the linear line occurred at around −log_10_(*P)* = 4.0, which presumably indicates true positives (Figure 2b–h). For RGR, −log_10_(*P*) values and QQ plots suggested relatively weak genetic association (Figure 2a). By applying the threshold of −log_10_(*P)* ≥ 4.0, the 2062 SNP markers covered all 13 chromosomes and 100 SNP markers that were unknown location (Table 3). Chromosome 3 had the maximum number of SNPs (332 SNPs), and Chromosome 12 had the minimum (57) number of SNPs. Among the nucleotide polymorphisms, 1708 SNPs were interginic, 96 SNPs were intronic, 68 SNPs were exonic, 112 SNPs were upstream, 69 SNPs were downstream and 9 SNPs were upstream and downstream (Appendix A).

Among these 2062 marker-trait associations, 61 markers were associated with RGR, 187 markers were associated with RFW, 255 markers were associated with RSL, 370 markers were associated with RWC, 190 markers were associated with RChlC, 583 markers were associated with REC, 335 markers were associated with RMDA and 81 markers were associated with CIST (Table 3, Appendix A) The most SNPs related positive salt tolerance indicators (RFW, RSL and RWC) were on chromosome 7. The most SNPs related negative salt tolerance indicators (REC, RMDA) were on chromosome 3 (Table 3).

### 2.3. SNP Rich Regions Associated with RFW and RSL

In the following, we focused on nine SNP rich regions associated RFW, RSL, RChlC, RWC and CIST for which MLMM analysis yield more significant associations considering −log_10_*P* ≥ 4.0 values and the position of strong peaks in the Manhattan plots (Appendix A).

Two SNP rich regions on chromosome 7 were found to associate with two biomass-related traits such as RFW and RSL (Figure 3a,b) when setting the threshold of −log_10_*P* ≥ 6.0 on the Manhattan plots. The first candidate region (Group 1) starting at 33,513,007 bp and ending at 33,616,148 bp (103,141 bp), on chromosome 7, which contained 6 polymorphism SNPs (Figure 3a,b) and located within five genes, Cotton_A_37774, Cotton_A_37775, Cotton_A_37776, Cotton_A_37777 and Cotton_A_40811. All six SNPs were related to RSL, and three SNPs were related to related fresh weight (RFW) (Appendix A). Two intronic SNPs (Ca7_33606785 and Ca7_33607751) related with both of RSL and RFW), including key SNP Ca7_33607751 (−log_10_(*P*) = 7.14, possession 33693051 bp) in this region, were in gene of Cotton_A_37775 (Appendix A Group 1). Four intergenic SNPs were related to related fresh weight (RFW) (Appendix A Group 1). Cotton_A_37775 was annotated as heat shock protein, a homolog of Arabidopsis thaliana AT5G52640. Haplotype analysis showed a low level of linkage disequilibrium (LD) (lowest *r2* = 30, highest *r2* = 100) between the associated SNPs in Group 1 (Figure 3c,d). There were three genotypes for the key SNP Ca7_33607751. Genotypes CC, TT and CT contained 7, 135, and 6 accessions respectively, whereas the accessions carrying CC genotype showed the highest RFW and RSL; the accessions carrying CT genotype showed medium-higher RWF (0.909) and RSL (0.775); And the accessions carrying TT genotype showed the lowest RFW (0.81) and RSL (0.698) (Figure 3e,f).

We then focused on the second highest peak (Group 2) on chromosome 7, which were common related to two biomass-related traits such as RFW and RSL. Group 2 was estimated to be 76,964,079 –77,073,963 bp (109,884 bp), and to contain 9 polymorphisms, which were located within 6 genes (Figure 3a,b and Appendix A). Among the 9 polymorphisms, 2 SNPs (Ca7_77000431 and Ca7_77004962) were related with both traits such as RFW and RSL, 7 SNPs were related only RSL. Most of these SNPs (6 of 9), including key SNP Ca7_77004962 (−log_10_(*P*) = 8.36) were located between two genes *Cotton_A_35901* and *Cotton_A_35900* (Appendix A). Haplotype analysis showed a low level of LD (lowest *r*^2^ = 15, highest *r*^2^ = 100) between the associated SNPs in Group 2 (Figure 3d). There were two genotypes for the key SNP Ca7_77004962. Genotypes AA and GG contained 8 and 146 accessions, respectively, whereas the accessions carrying AA genotype showed the highest RFW (1) and RSL (1); the accessions carrying GG genotype showed lower RWF (0.808) and RSL (0.697) (Figure 3g,h). 

### 2.4. SNP Rich Regions Associated with RChlC

Similarly, we analyzed four SNP rich regions on chromosome 4, 9 and 11, which closely associated with RChlC (Figure 4a). The candidate region on chromosome 4 (Group 3) was predicted to map from 32,124,549 bp to 32,259,755 bp (135,206 bp), and contained 14 polymorphisms, which were located within 6 genes (Figure 4a, Appendix A and Appendix A Group 3). Eight SNPs were distributed near the gene of *Cotton_A_26219* (Appendix A). We found that two haplotypes in three SNPs (Ca4_32185154, Ca4_32191704 and Ca4_32197265) by using high pairwise LD correlation (*r*^2^ ≥ 96) (Figure 4b,f). These three SNPs were near *Cotton_A_26219*. Haplotype A and Haplotype B contained 27 and 157 accessions respectively (Figure 4f). The accessions carrying haplotype A showed lower RChlC (0.80) than haplotype B (0.97) (Figure 4i).

We estimated the candidate region on chromosome 9 (Group 4) to be 56,869,491–56,879,931 bp (10,440 bp) and assigned 14 polymorphisms, which were located within 3 genes (Figure 4a, Appendix A and Appendix A Group 4). Most of these polymorphisms (8 of 14), including the key SNP Ca9_56878752 (exonic) were in *Cotton_A_10864*. Five coding region SNPs in *Cotton_A_10865*, which is annotated as F-Box protein and leucine-rich repeat protein 14 (Appendix A Group 4). We found that two haplotypes in seven SNPs (Ca9_56875436, Ca9-56875607, Ca9-56876686, Ca9-56876694, Ca9-56877663 in Cotton_A_10865, and Ca9-56878677, Ca9-56878752 in *Cotton_A_10864*) by using high pairwise LD correlation (*r*^2^ > 80) (Figure 4c,g). Haplotype A and Haplotype B contained 9 and 84 accessions, respectively (Figure 4g). The accessions carrying haplotype A showed higher RChlC (1) than haplotype B (0.93) (Figure 4j).

The candidate region on chromosome 9 (Group 5) was predicted to map from 92,702,808 bp to 92,735,184 bp (32,376 bp), and contained 19 polymorphisms (Figure 4a, Appendix A and Appendix A Group 5). All 19 significant SNP markers (key SNP Ca9_92711930, −log_10_*P* = 5.80) was located between two pathogenesis-related thaumatin superfamily protein genes: *Cotton_A_15275* (average distance 13,817 bp) and *Cotton_A_15276* (average distance 31,662 bp) (Appendix A Group 5). We found that two haplotypes in thirteen SNPs (from Ca9_92710379 to Ca9_92717732) by using high pairwise LD correlation (*r*^2^ ≥ 91) (Figure 4d,h). Haplotype A and Haplotype B contained 12 and 97 accessions, respectively (Figure 4h). The accessions carrying haplotype A showed higher RChlC (1) than haplotype B (0.92) (Figure 4k).

The candidate region on chromosome 11 (Group 6) was predicted to map from 47,006,642 bp to 47,011,718 bp (5076 bp), and contained 11 polymorphisms (Figure 5a, Appendix A and Appendix A Group 6). All 11 significant SNP markers (key SNP Ca11_47011718, −log_10_*P* ≥ 5.61) was located between two genes: *Cotton_A_28249* (average distance from SNPs 7649 bp) and *Cotton_A_28248* (average distance 5083 bp from SNPs). There were three genotypes for the key Ca11_47011718 (Appendix A Group 6). Genotypes AA, GG and AG contained 190, 13, and 2 accessions respectively, whereas the accessions carrying AA (0.933) genotype showed lower RChlC; And the accessions carrying GG (1) and AG (1) genotype showed the highest RChlC (Figure 5i).

### 2.5. SNP Rich Region Associated with RWC

The candidate region on chromosome 7 (Group 7) was predicted to map from 39,823,916 bp to 39,843,478 bp (19,562 bp), and contained 29 polymorphisms in four genes (*Cotton_A_05854*, *Cotton_A_05853*, *Cotton_A_05852* and *Cotton_A_05852*) (Figure 5a, Appendix A and Appendix A Group 7). Among these four genes, most important gene is *Cotton_A_05853*, because, 15 of these SNPs (11 intronic and 3 intergenic), including intronic key SNP Ca7_39832729 (−log_10_(*P*) > 5.35) were located in *Cotton_A_05853* (*AT3G12800*), annotated in The Arabidopsis Information Resource (TAIR) as short-chain dehydrogenase-reductase B and oxidation-reduction process (Appendix A Group 7). We found that two haplotypes of eight SNPs in *Cotton_A_05853* (from Ca7_39832407 to Ca7_39832920) by using high pairwise LD correlation (*r*^2^ ≥ 90) (Figure 5b). Haplotype A and Haplotype B contained 42 and 6 accessions, respectively (Figure 5c). The accessions carrying haplotype A showed higher RWC (0.86) than haplotype B (0.50) (Figure 5d). 

### 2.6. SNP Rich Regions Associated with CIST

The candidate region on chromosome 2 (Group 8) was predicted to map from 86,947,129 to 87,017,559 bp (11,697 bp) and to contain 16 polymorphisms in two genes (*Cotton_A_22673*, *Cotton_A_22672*) (Figure 6a, Appendix A and Appendix A Group 8). Fifteen significant intergenic SNPs, including key SNP Ca2_86954790 (−log_10_(*P*) ≥ 6.32) were located near the gene of *Cotton_A_22673* (average distance 63,317 bp). Only one significant non-synonymous SNP is in *Cotton_A_22672*, which without annotation (Appendix A Group 8). We found that two haplotypes of six SNPs near to *Cotton_A_22673* (from Ca2_86947129 to Ca2_86956247) by using high pairwise LD correlation (*r*^2^ ≥ 93) (Figure 6b). Haplotype A and Haplotype B contained 27 and 157 accessions, respectively (Figure 6d). The accessions carrying haplotype A showed lower CIST (1.042) than haplotype B (1.935) (Figure 6f).

The candidate region on chromosome 11 (Group 9) was predicted to map from 39,493,066 to 39,504,763 bp (11,697 bp) and to contain 9 polymorphisms in two genes (*Cotton_A_21725* and *Cotton_A_21726*) (Figure 6a, Appendix A and Appendix A Group 9). *Cotton_A_21726* contained eight significant intronic SNPs, including key SNP Ca11_39504708 (−log_10_*P* = 5.98), annotated Glycosyl hydrolase family 10 proteins. Only one significant downstream SNP was in *Cotton_A_21725*, annotated as DNA/RNA polymerases superfamily protein (Appendix A Group 9). We found that three haplotypes in all nine SNPs in this region by using very high pairwise LD correlation (*r*^2^ = 100) (Figure 6c,e). Haplotype A, Haplotype B and Haplotype C contained 35, 5 and 36 accessions, respectively (Figure 6e). The accessions carrying haplotype A (1.973) and C (2.056) showed similar CIST, but the accessions carrying haplotype B showed the lowest CIST (−0.1398) (Figure 6g).

## 3. Discussion

### 3.1. Genetic Variation in Salt Tolerance Related Traits of G. arboretum Accessions 

*G. arboretum* is considered superior to upland cotton varieties based on the following traits: precociousness, wide adaptability, drought tolerance and disease resistance from fusarium wilt, insects and bell mild disease [32]. However, it has not been well characterized at the molecular level. Thus, our current study mainly focused on identification and screening of salt tolerant germplasm during seedling stage, to find genetic relationships and to study marker-trait associations SNP markers. The *G. arboretum* accessions, considered as an invaluable gene pool for cotton improvement, were used in this study. Understanding of the genetic diversity of *G. arboretum* can facilitate the efficient use of these resources in the development of superior cotton cultivars with favorable agronomic traits. 

Abiotic stress leads to a series of morphological, physiological, biochemical and molecular changes that have adverse effects on the plant growth, development and productivity. In fact, the salinity is a major abiotic stress that limits cotton growth and development at the germination and seedling stages [27]. In this study, the suitable optimum NaCl concentration was determined by monitoring germination rates of accessions. Shixiya No. 1, Sichuansuining and Chaoxianjinhuaxiaozi (included in the 215 accessions) under different NaCl concentrations (0, 50, 100, 150, 200, 250, and 300 mM). Under 200, 250, and 300 mM NaCl treatments, most seeds were unable to geminate. In addition, most of the seeds germinated under 100 and 150 mM NaCl treatments (Appendix A). Based on these results and the reports of Chen et al. (2008), 150 mM NaCl concentration was considered suitable salt treatment for *G. arboretum* [29]. Under 150 mM NaCl treatment, significant differences among different cotton accessions were observed during germination and seeding stages. The assessment of diversity in a species is important in plant breeding programs, and for effective conservation, management and utilization of genetic resources of the species [30]. The salt tolerance related traits (GR, FW, SL, WC, ChlC, EC, MDA) were comparable to that of the salt tolerance study and reported previous works [33]. Notably, plant germination rate, stem length, fresh weight are the major components of plant yield and were used as selection criteria in breeding. The ANOVA for the important salt tolerance parameters revealed significant differences (*P* < 0.0001) among the genotypes, implying that sufficient phenotypic polymorphism existed between individual *G. arboretum* accessions in this study. The correlation analysis is important to identify the mutual associations among the traits [34]. There was efficiency correlation between seven salt-tolerance-related traits; it is useful for multiple trait selection at one time for development of improved cotton varieties. The classification information derived from these studies may be used to facilitate the development of salt tolerant cotton accessions that could give economic yield in salinity prone areas.

### 3.2. Association Mapping of Salt Tolerance Traits Using SNP Markers

Genome-wide association study (GWAS) can effectively associate genotypes with phenotypes in natural populations and simultaneously detect many natural allelic variations and candidate genes in a single study, in contrast to QTL linkage mapping [35]. To cope with environmental stress, plants activate a large set of genes leading to the accumulation of specific stress-associated proteins. In this study, we first performed a genome-wide association analysis of salt tolerance related traits with 215 of natural accessions of *G. arboretum*. This study uncovered 2062 loci (−log_10_*P* ≥ 4.0) associated salt tolerance traits and identified a set of candidate genes that could be exploited to alter salt tolerance development to improve *G. arboretum* accessions. The analysis of genomic distribution of SNPs in this study revealed that the most of SNPs related positive salt tolerance indicators (RFW, RSL and RWC) were on chromosome 7 and the most of SNPs related negative salt tolerance indicators (REC, RMDA) were on chromosome 3. 

The nine SNP rich regions analysis revealed 143 polymorphisms that distributed 40 candidate genes and significantly associated RFW, RSL, RChlC, RWC and CIST. We found that twelve and eight SNPs in Group 1 and Group 2 were associated with two biomass traits such as RFW and RSL, because there is highly significant correlation (*P* < 0.001) between FW and SL (Appendix A Group 1 and Group 2).

In the first SNP rich region (Group 1), the most plausible candidate indented in the peak on chromosome 7 was *Cotton_A_37775* that was involved in Heat shock protein (Hsps), which plays a crucial role in salt stress response. *Cotton_A_37775* contains 1,951 amino acid and shares 79% identify at the amino acid level with a homolog of *Arabidopsis thaliana AT5G52640*, which annotated Heat shock protein (Hsp90.1). The 90 kDa heat shock protein (Hsp90) is a widespread family of molecular chaperones found in prokaryotes and all eukaryotes [36]. Hsp90 chaperone machinery is a key regulator of proteostasis under both physiological and stress conditions in eukaryotic cells. A large number of co-chaperones interact with HSP90 and regulate the ATPase-associated conformational changes of the HSP90 dimer that occur during the processing of clients [37]. The basic functions of HSP90s are of assisting protein folding, protein degradation and protein trafficking, and they also play an important role in signal transduction networks, cell cycle control and morphological evolution. Although HSP90s are constitutively expressed in most organisms, their expression is up-regulated in response to stresses such as cold, heat, salt stress, heavy metals, phytohormones, light and dark transitions [38]. The most prominent response of plants under high temperature stress is the rapid production of heat shock proteins (HSPs). Ding (2006) found that heat shock treatment in early growing period benefited the abundance of HSPs in cotton leaves in high temperature season, and then increased the ability thermo-tolerance [39]. In the plant *Arabidopsis thaliana (A. thaliana)*, *HSP90* homologs are encoded by seven different genetic loci. Of these, one is expressed in the endoplasmic reticulum (*HSP90.7*), one in the mitochondrion (*HSP90.6*), one in the chloroplast (*HSP90.5*), and four in the cytosol. The gene encoding one cytosolic protein (*HSP90.1*/*At5g52640*) is highly stress-inducible, whereas the other three (*HSP90.2*/*At5g56030*, *HSP90.3*/*At5g56010*, and *HSP90.4*/*At5g56000*) are constitutively expressed and are the products of very recent duplication events [40]. The homolog *At5g52640* is found up-regulated in response to viruses stresses by playing a role in cell migration [41]. 

In the second SNP rich region (Group 2), we identified a gene, *Cotton_A_35901* (301 amino acid), encoding a SNARE-like superfamily protein homologue, which has not been previously reported in cotton. Complexes of SNARE proteins mediate intracellular membrane fusion between vesicles and organelles to facilitate transport cargo proteins in plant cells [36] 

In the third SNP rich region (Group 3), we found that two haplotypes in three SNPs (Ca4_32185154, Ca4_32191704 and Ca4_32197265) by using high pairwise LD correlation (r^2^ ≥ 96). These three intergenic haplotype SNPs located between two genes *Cotton_A_26219* and *Cotton_A_26218*. *Cotton_A_26219* have no annotation. Anther gene *Cotton_A_26218* is a homologue of *A.thaliana AT1G18800*, which encoding nucleosome assembly protein (nrp1-1 nrp2-1). Juan (2012) show that the nucleosome assembly protein (NAP1) family histone chaperones are required for somatic homologous recombination (HR) in *A. thaliana* [42]. HR is essential for maintaining genome integrity and variability. To orchestrate HR in the context of chromatin is a challenge, both in terms of DNA accessibility and restoration of chromatin organization after DNA repair. Histone chaperones function in nucleosome assembly/disassembly and could play a role in HR.

Depletion of either the NAP1 group or NAP1-RELATED PROTEIN (NRP) group proteins caused a reduction in HR in plants under normal growth conditions as well as under a wide range of genotoxic or abiotic stresses. In *Arabidopsis thaliana*, *AT1G18800* is required for maintaining cell proliferation and cellular organization in root tips [43].

The analysis in the fourth SNP rich region (Group 4) revealed two haplotypes in seven SNPs distributed in two candidate genes, Cotton_A_10865 and Cotton_A_10864 on Chr9, were associated with RChlC. Cotton_A_10865 is homologous to *AT4G08980* (F-BOX WITH WD-40), encodes an F-box gene that is a novel negative regulator of AGO1 protein levels and may play a role in abscisic aci (ABA) signaling and/or response. ABA signaling also plays a major role in mediating physiological responses to environmental stresses such as salt, osmotic, and cold stress. The accumulation of ABA in response to water or salt stress is a cell signaling process, encompassing initial stress signal perception, cellular signal transduction and regulation of expression of genes encoding key enzymes in ABA biosynthesis and catabolism [44]. In addition, there are a lot of studies to prove ABA response to salt stress in different crops such as *A. thaliana* [44,45,46,47], rice [48], wheat [49], corn [50]. The homologue of *Cotton_A_10864* is *AT5G65270* (RAB GTPASE HOMOLOG A4A) in *A. thaliana*. Several genes in the Rab GTPase family have been shown to be responsive to abiotic stress, including response of *SsRab2* to water stress in *Sporobolus stapfianus* [51], *MfARL1* to salt stress in *A. thaliana* [47], OsRab7 to cold stress in rice [52] and *AtRabG3e* to salt/osmotic stress in *A. thaliana* [53].

In the fifth SNP rich region (Group 5), the two genes (*Cotton_A_15275* and *Cotton_A_15276*) in Group 5 were pathogenesis-related thaumatin superfamily protein. Pathogenesis-related (PR) proteins play an important role in plants as a protein-based defensive system against abiotic and biotic stress, particularly pathogen infections. It is also named thaumatin-like proteins (TLPs), because it has sequence similarity with thaumatin [54,55]. 

In the six SNP rich region (Group 6), we found two genes *Cotton_A_28248* and *Cotton_A_28249*. *Cotton_A_28248* has no annotation. *Cotton_A_28249* is homologous to *AT4G32050* and encodes neurochondrin family protein, which is an atypical RIIα-specific A-kinase anchoring protein. In the seventh SNP rich region (Group 7), we found two haplotypes in seven SNPs distributed in one gene, *Cotton_A_05853*, has no annotation. 

In the eighth SNP rich region (Group 8), we found two candidate genes *Cotton_A_22673* and *Cotton_A_22672*. *Cotton_A_22673* is annotated carbon-sulfur lyases, homologous to *A.thaliana AT5G09970*. *AT5G09970* locate in mitochondrion and encodes a DYW-class PPR protein required for RNA editing at four sites in mitochondria of *A. thaliana*. 

In the ninth SNP rich region (Group 9), we found two candidate genes *Cotton_A_21725* and *Cotton_A_21726*. *Cotton_A_21725* encodes DNA/RNA polymerases superfamily protein. Several genes are induced under the influence of various abiotic stresses. Among these are DNA repair genes, which are induced in response to the DNA damage. Since the stresses affect the cellular gene expression machinery, it is possible that molecules involved in nucleic acid metabolism including helicases are likely to be affected. The light-driven shifts in redox-potential can also initiate the helicase gene expression. Helicases are ubiquitous enzymes that catalyze the unwinding of energetically stable duplex DNA (DNA helicases) or duplex RNA secondary structures (RNA helicases). Most helicases are members of DEAD-box protein superfamily and play essential roles in basic cellular processes such as DNA replication, repair, recombination, transcription, ribosome biogenesis and translation initiation. Therefore, helicases might be playing an important role in regulating plant growth and development under stress conditions by regulating some stress-induced pathways [56]. *Cotton_A_21726* encodes glycoside hydrolasefamily 10. GHs (glycosyl hydrolases) enzymes that catalyze the hydrolysis of glycosidic bonds between sugars and other moieties, can be classified into more than 100 families [55]. We could not find any relationship between glycoside hydrolase family 10-prtain and salt tolerance. But the glycoside hydrolase family 5 gene is expressed in rice leaves and seedling shoots, whereas its expression is induced by stress-related hormones, submergence and salt in whole seedlings [57]. 

The identified genetic variation and candidate genes deepen our understanding of the molecular mechanisms underlying salt tolerance traits. The genes discussed in this study may be considered as candidate genes for salt tolerance in cotton.

## 4. Materials and Methods

### 4.1. Plant Materials and Sample Preparation

The genetic materials used in the present study includes 215 accessions of *G*. *arboretum*, among them 209 accessions belong to China, 3 accessions from the United States, and 3 accessions from India, Pakistan and Japan. The germplasm was assembled from Germplasm bank of Institute of Cotton Research of Chinese Academy of Agricultural Sciences (CAAS), Anyang, China. The detailed list of accessions along with their origin is described in Appendix A.

The phenotypic analysis and genetic experiment for these selected accessions were performed in the laboratory of Cotton Research Institute of CAAS, Anyang, China. Seedlings were grown in a phytotron incubating chamber under 14 h/10 h (light/dark) cycle, 28/14 °C (day/night), 450 μmol·m^−2^⋅s light intensity and a relative humidity of 60–80% conditions [34]. For each genotype, 200 hand-selected seeds for each variety were sterilized with 15% hydrogen peroxide for 4 h, and subsequently rinsed with sterile distilled water at least 4 times, followed by seed submersion in sterile water for 12 h at room temperature. For identification of the seed germination rates (GR), 120 healthy seeds from each accession were selected and placed in germination boxes (200 × 150 mm diameter), containing two sheets of filter paper and soaked with 20 mL of the NaCl solutions (0 or 150 mM) respectively. For identification of other traits, such as fresh weight (FW), stem length (SL), water content (WC), chlorophyll content (ChlC), electric conductivity (EC) and methylene dioxyamphetamine (MDA), twenty-five healthy seeds each were planted in the 300 mL volumetric flask (Appendix A), containing 140 g sand (sterilized in autoclave at 160 °C for 2 h), 40 mL sterilized water (for control 0 mM) and 40 mL NaCl solutions (for 150 mM). 

### 4.2. Trait Evaluation

After seven days of the germination, the 215 *G. arboretum* accession seedlings grown in the soil were evaluated for seven salt-tolerance-related traits, such as GR, FW, SL, WC, ChlC, EC and MDA. The seeds were considered to have germinated when the radicle and plumule length was equivalent to the seed length or half of the seed length. The germination rate was calculated GR (%) = (number of germinated seeds/total seed number used in the test) × 100. Fifteen plants from each accession were harvested and cleaned by sterilized water and immediately used for determination of FW, SL and ChlC.

For leave water content (WC) estimation, detached leaves were floated on deionized water for 8 h at 4 °C and turgid weights (TW) were determined. Later, dry weight (DW) of leaves was measured after oven-dried at 105 °C for 10 min, and then 80 °C for 24 h respectively. WC was calculated WC (%) = (FW − DW)/(TW − DW) × 100 [33].

For EC measurement, 0.5 g of leaves were rinsed with double distilled water (ddH_2_O) and put in volumetric flask containing 40 mL of ddH_2_O. Afterwards, the test flasks were incubated at room temperature for 4 h. The electrical conductivity of the solution (C1) was measured using a conductivity meter EM38 (ICT, Australian). The test flasks were boiled for 10min and then cooled at room temperature to measure the electrical conductivity (C2). The REC was calculated using the formula C1/C2 × 100%. [33]

For MDA measurement, 0.5 g of leaves was rinsed with double distilled water (ddH_2_O) and then crushed in Thiobarbituric acid extract solution (5 mL, 0.5%). Absorbances were monitored at three different wavelengths i.e., 450 nm, 532 nm and 600 nm. MDA was calculated according to Le et al. (2000) method. MDA (X) = [6.452*(OD532 − OD600 − 0.559*OD450] *Vt/(Fw × 1000) [33]. Where X is MDA in µmol/g, Vt is total volume of extraction solution in mL and FW is fresh weighting.

All phenotypic data and physiological indicators were performed at least three biological replicates. The relative value of GR phenotypic data was used for further GWAS. The formula is RGR = GR_150_/GR_control_, i.e., the same as that for other traits, RFW, RSL, RWC, RChlC, REC and RMDA.

### 4.3. DNA Extraction

Young leaves were collected from five plants of each accession and stored at −80 °C. The DNA was isolated from the frozen leave using CTAB method [34] with some modifications. DNA concentration was quantified using a NanoDrop2000 instrument (Thermo Scientific, USA), and normalized to 50 ng/mL.

### 4.4. Phenotypic Diversity

Analysis of variance (ANOVA) and phenotypic correlations between different salt related physiological traits were performed using statistical software package SAS 9.21. Relative value for each trait and CIST were calculated according to these formulae:

Relative value = value under stress treatment (S)/value under control treatment (C); 

CIST = positive index (RGR + RSL + RFW + RChlC + RWC)/negative index (REC + RMDA).

### 4.5. Genome-Wide Association Analysis 

The SNP markers associated with the seven traits and one salt tolerance index were identified using the MLMM model in the EMMAX software [58]. Key SNP analysis was identified by IGV_2.3.83_4 (Integrative genomic viewer, version 2.3.83_4). Lastly, ~1.57 million high-quality SNPs (MAF > 0.05, missing rate < 40%) were obtained and used for the further GWAS. The genome-wide significance thresholds of all tested traits were evaluated using −log_10_(*P*) ≥ 4.0.

### 4.6. Statistical Analysis

For phenotypic experiment, each data represented the mean of three or more biological replicate treatments, with each treatment consisting of at least five plants. The statistical analyses were performed using Tukey’s two-way analysis of variance (ANOVA) in IBM SPSS Statistics v19.0 (SPSS Inc., Chicago, IL, USA). *P*-values < 0.001 were considered statistically significant. Pearson correlation also was used SPSS V19.0. The different star between two traits represent significance at *P*-value < 0.05, 0.01, 0.001.

## 5. Conclusions

We firstly investigated the genetic architecture of natural variation in Asian cotton salt-tolerance-related traits at the seedlings stage by GWAS mapping in 215 accessions. The SNP markers and candidate genes in this study may be used as references for other association mapping studies of salt tolerance. The salt tolerance related novel candidate genes will provide an important resource for molecular breeding and functional analysis of the salt tolerance during the cotton germination.

## Figures and Tables

**Figure 1 ijms-20-02168-f001:**
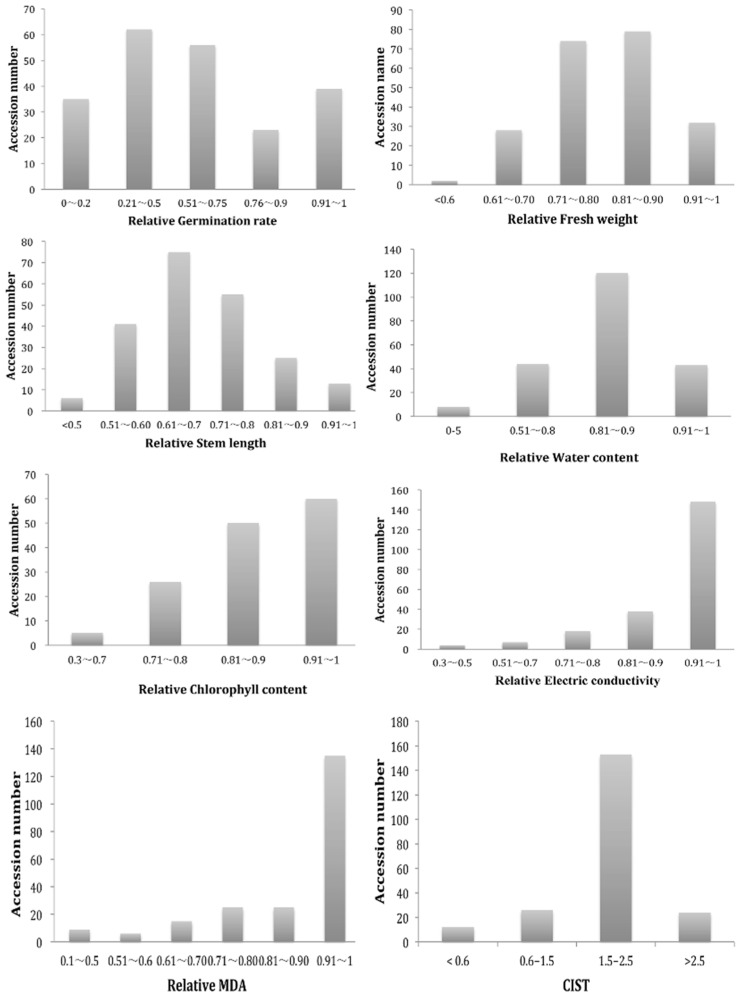
Relative value frequency distribution diagram of seven salt tolerance traits and one salt tolerance index of 215 *G. arboreum* accessions.

**Figure 2 ijms-20-02168-f002:**
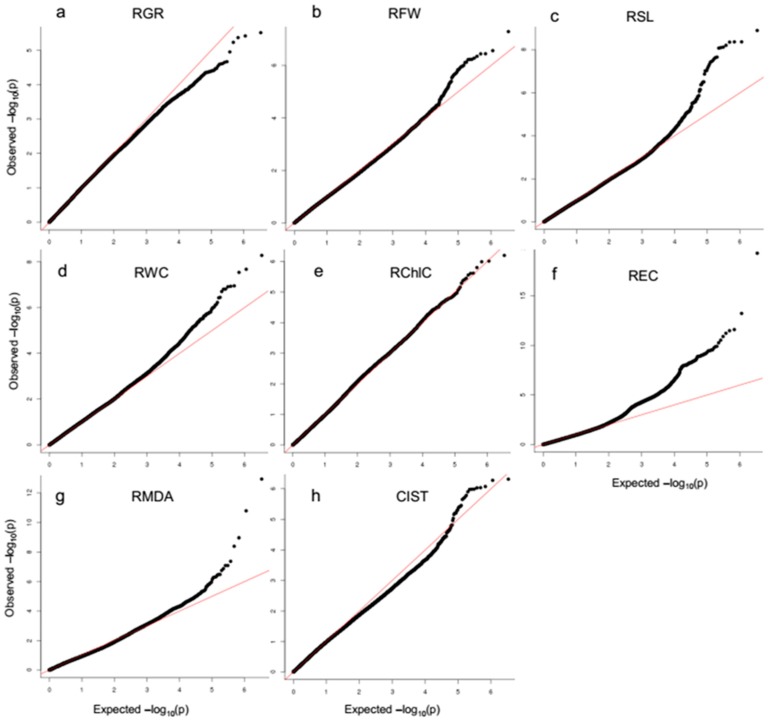
Quantile-quantile plots of versus expected −log_10_*P* values of GWAS result. The red dashed line in each plot represents an idealized case where theoretical test statistics quantile match simulated test statistic quantile. (**a**) Relative germination rate (RGR). (**b**) Relative fresh weight (RFW). (**c**) Relative stem length (RSL). (**d**) Relative water content (RWC). (**e**) Relative chlorophyll content (RChlC). (**f**) Relative electric conductivity (REC). (**g**) Relative MDA (RMDA). (**h**) Comprehensive index of salt tolerance (CIST).

**Figure 3 ijms-20-02168-f003:**
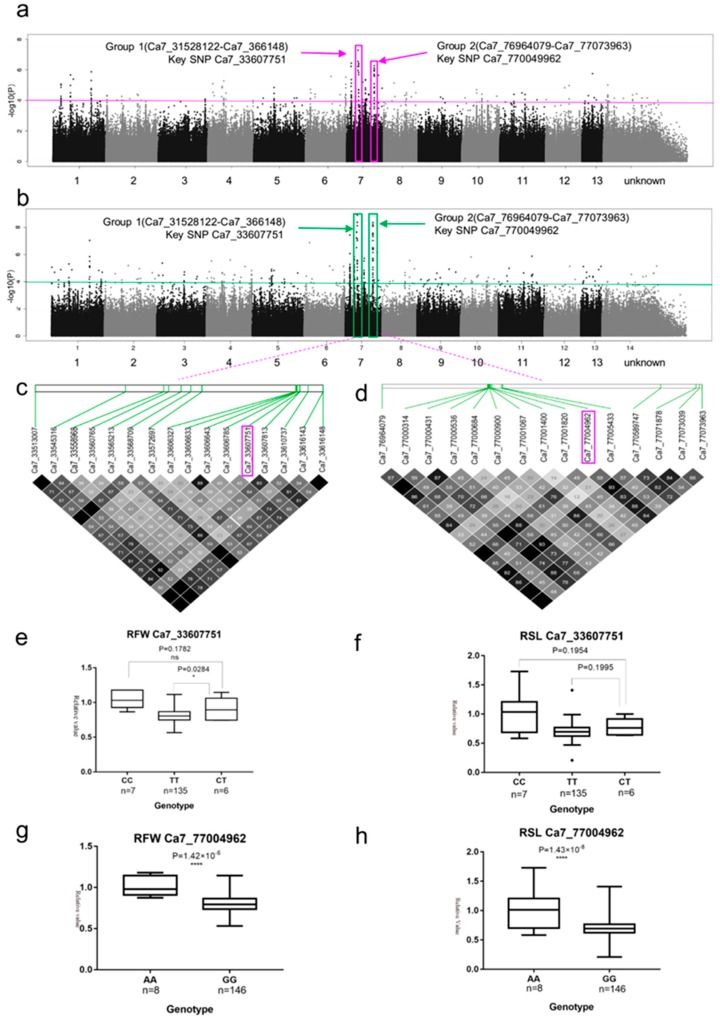
GWAS results for RFW and RSL, and analysis of the peaks on chromosome 7 (Group 1 and Group 2). (**a**) Manhattan plot for RFW. The horizontal line represents the significant threshold (−log_10_*P* = 4). The pink color surrounds represent SNP rich regions (Group 1 and Group 2). (**b**) Manhattan plot for RSL. The horizontal line represents the significant threshold (−log_10_*P* = 4). The pink color surrounds represent SNP rich regions (Group 1 and Group 2). (**c**) LD surrounding the peak of Group 1. (**d**) LD surrounding the peak of Group 2. (**e**) Phenotypic differences for RFW based on the key SNP (Ca7_33607751) of Group 1. (**f**) Phenotypic differences for RSL based on the key SNP (Ca7_33607751) of Group 1. (**g**) Phenotypic differences for RFW based on the key SNP (Ca7_77004962) of Group 2. (**h**) Phenotypic differences for RSL based on the key SNP (Ca7_77004962) of Group 2.

**Figure 4 ijms-20-02168-f004:**
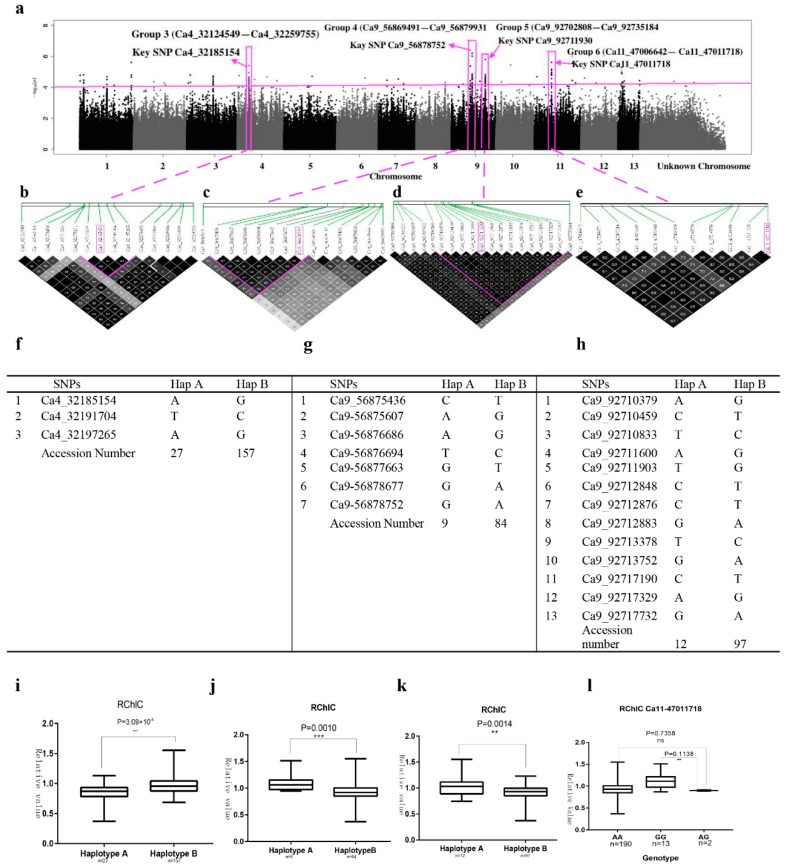
GWAS results for RChlC and analysis of the peaks on chromosome 4, 9 and 11. (**a**) Manhattan plot for RChlC. The horizontal line represents the significant threshold (−log_10_*P* = 4). The pink color surrounds represent SNP rich regions (Group 3, Group 4, Group 5 and Goup 6). (**b**) LD surrounding the peak of Group 3. (**c**) LD surrounding the peak of Group 4. (**d**) LD surrounding the peak of Group 5. (**e**) LD surrounding the peak of Group 6. (**f**) Haplotypes in Group 3. (**g**) Haplotypes in Group 4 (**h**) Haplotypes in Group 5. (**i**) Phenotypic differences of RChlC between two haplotypes in Group 3. (**j**) Phenotypic differences of RChlC between two haplotypesin Group 4. (**k**) Phenotypic differences of RChlC between two haplotypesin Group 5. (**l**) Phenotypic differences for RChlC based on the key SNP (Ca11_47011718) of Group 6.

**Figure 5 ijms-20-02168-f005:**
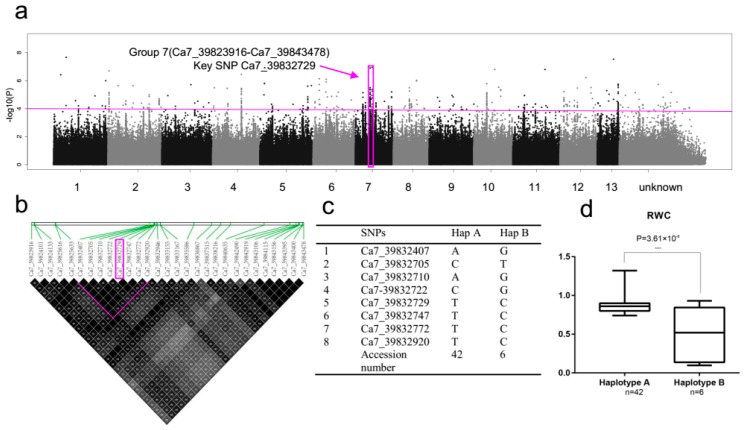
GWAS results for RWC and analysis of the peak on chromosome 7. (**a**) Manhattan plot for RWC. The horizontal line represents the significant threshold (−log_10_*P* = 4). The pink color surround represents SNP rich region (Group 7). (**b**) LD surrounding the peak of Group 7. (**c**) Haplotypes in Group 7. (**d**) Phenotypic differences of RWC between two haplotypes.

**Figure 6 ijms-20-02168-f006:**
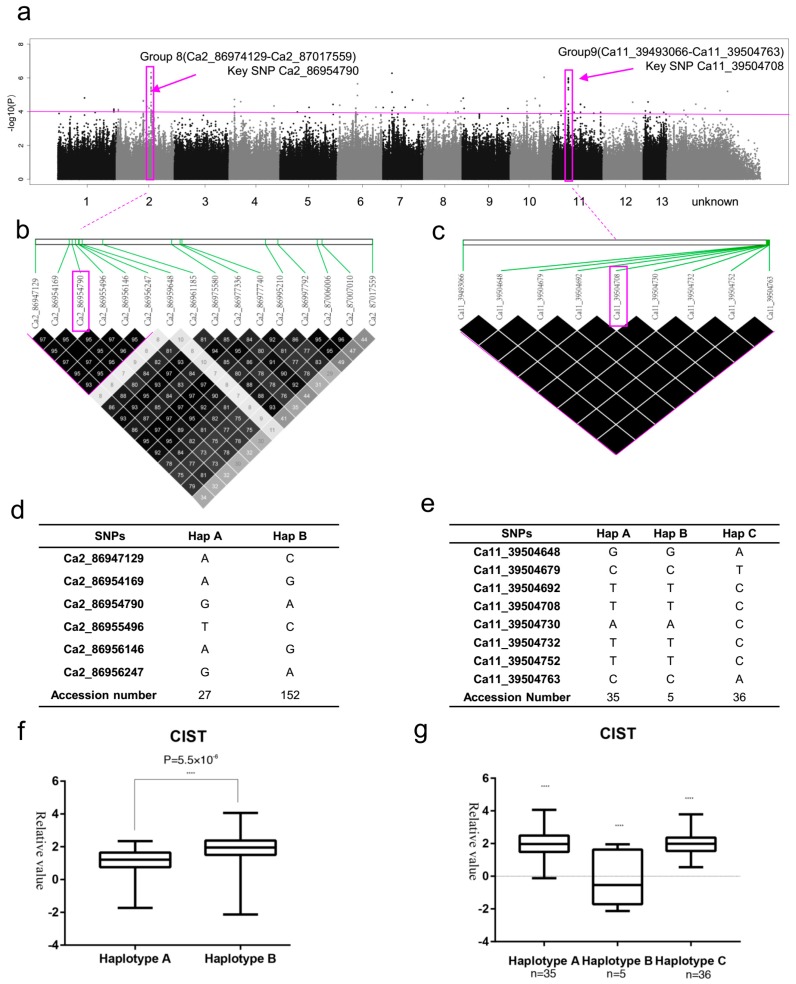
GWAS results for CIST and analysis of the peaks on chromosome 2 and 11. (**a**) Manhattan plot for CIST. The horizontal line represents the significant threshold (−log_10_*P* = 4). The pink color surrounds represent SNP rich regions (Group 8 and Group 9). (**b**) LD surrounding the peak of Group 8. (**c**) LD surrounding the peak of Group 9. (**d**) Haplotypes in Group 8. (**e**) Haplotypes in Group 9. (**f**) Phenotypic differences of CIST between two haplotypes in Group 8. (**g**) Phenotypic differences of CIST among three haplotypes in Group 9.

**Table 1 ijms-20-02168-t001:** Analysis of the traits related salt treatment in *G. arboretum* accessions.

Traits ^1^	Mean	SD	Min	Max	CV	Mean Square	F	P > F
**GR**	19.936	10.77	0	40	54.02	338.55	8.16	<0.0001
**FW**	0.371	0.089	0.02	0.77	24.2	0.128	34.16	<0.0001
**SL**	4.969	1.481	0.40	10.2	29.84	40.77	49.76	<0.0001
**WC**	1.112	3.481	0.36	96.97	312.9	31.83	4.01	<0.0001
**ChlC**	39.47	9.96	3.50	50.2	25.53	6763.5	8.31	<0.0001
**EC**	31.47	42.54	0.116	844.4	135.19	541.2	6.49	<0.0001
**MDA**	0.007	0.0046	0	0.04	59.8	0.000095	14.36	<0.0001

^1^
*GR* germination rate; *FW* fresh weight; *SL* stem length; *WC* water content; *ChlC* relative chlorophyll content; *EC* electric conduct; *MDA* methylene dioxyamphetamine.

**Table 2 ijms-20-02168-t002:** Analysis of traits related salt treatment of *G. arboretum* accessions (Pearson correlation coefficient).

Trait ^1^	GR	FW	SL	WC	ChlC	EC	MDA
**GR**	1	0.22 ***	0.295 ***	−0.0178	0.112 **	0.0841	0.012
**FW**		1	0.575 ***	−0.043	0.070 ***	0.0613	0.135 ***
**SL**			1	−0.017	0.115 ***	−0.051	0.030
**WC**				1	0.048	0.002	0.059 *
**ChlC**					1	0.1 **	−0.073 *
**EC**						1	0.079 *
**MDA**							1

^1^ For trait abb. Look at Table 1. * Significant at *P* < 0.05; ** Significant at *P* < 0.01; *** Significant at *P* < 0.001 for the correlation coefficient.

**Table 3 ijms-20-02168-t003:** Associated SNPs of different salinity traits distribution on chromosome ^1^^.^

Chromosome	Total	RGR	RFW	RSL	RWC	RChlC	REC	RMDA	CIST
**Chr-1**	163	1	26	22	12	15	14	70	3
**Chr-2**	152	5	10	10	39	3	12	49	24
**Chr-3**	332	1	3	8	7	11	278	24	0
**Chr-4**	201	3	12	26	27	51	59	19	4
**Chr-5**	155	2	7	17	21	4	94	8	2
**Chr-6**	112	10	16	11	22	2	21	23	7
**Chr-7**	295	2	67	108	67	6	24	14	7
**Chr-8**	99	15	10	8	27	4	9	24	2
**Chr-9**	104	4	2	8	9	47	7	24	3
**Chr-10**	80	1	7	10	34	4	11	7	6
**Chr-11**	112	5	5	8	17	20	24	18	15
**Chr-12**	57	8	4	2	31	3	5	3	1
**Chr-13**	100	1	4	10	29	16	6	32	2
**Chr-UN**	100	3	14	7	28	4	19	20	5
**Total**	**2062**	**61**	**187**	**255**	**370**	**190**	**583**	**335**	**81**

^1^ the SNP exceeding a significant threshold(−log_10_(*P*) ≥ 4.0); 2 For trait abb. Look at Table 1; CIST comprehensive index of salt tolerance.

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
