# Peer review of "Association Analysis of Salt Tolerance in Asiatic cotton (Gossypium arboretum) with SNP Markers"

_ijms, 2019, doi:10.3390/ijms20092168_

Round 1

Reviewer 1 Report

The manuscript “Association Analysis of Salt Tolerance in Asiatic cotton (Gossypium arboretum) with SNP Markers” is the first research article to map SNP association related to salt tolerance in G. arboretum. Those markers may be used in cotton breeding with MAS.

The topic is of interest and the manuscript is well written. The English language is appropriate and understandable.

The authors provide a complete GWAS on salt responses in G. arboretum, and in my opinion only some minor aspects should be considered before acceptance for publication.

I recommend authors write a sentence about the genome size 1.7G(with 2n=26) of G. arboretum in the Introduction. That information will benefit SNP or SSR markers in GWAS and MAS.

Line 37, HCO3- should be HCO3- .

Question 1:

Line 21, 24, 114 and 140: authors use -log10(P)>4.0, why in Line 115, 119, 144 use -log10P≥4.0 ?

Question 2:

Line 166: authors said “The horizontal line represents the significant threshold (-log10P=4)”. But in the Figure 4a and 4b, the horizontal line looks like -log10P=4.2? Same question in Figure 6a.

Question 3:

In 3.2, authors find some candidate genes such as Cotton_A_37775, could they do some qRT-PCR to see whether those genes’ expressions are related to salt stress or not?

Question 4:

Several papers should be considered in the Introduction, as they are directly related with association mapping in G. arboretum.

Kantartzi S K, Stewart J M. Association analysis of fibre traits in Gossypium arboreum accessions. Plant Breeding, 2010, 127(2):173-179.

Cai C, Wu S, Niu E, Cheng C, Guo W. Identification of genes related to salt stress tolerance using intron-length polymorphic markers, association mapping and virus-induced gene silencing in cotton. Scientific Reports, 2017,7(1):528.

Author Response

The reply could be found in the attachment.

Reviewer 2 Report

This original work is very interesting and identified marker-trait association signals of seven salt-tolerance-related traits and one salt tolerance index using association analysis for 215 accessions of Asiatic cotton. However, the manuscript still needs some major improvements. I would suggest improving the manuscript title to better suit the main objective and findings ease. Additionally, the authors have to address the following raised points;

- Abstract
The abstract should be improved to include the main findings of important analyses carried out in this manuscript.

English grammar should be improved in the abstract

- Introduction
The introduction is somewhat well-written, but not covered the main objectives well. The introduction should also discuss the literature on such topic as well. Additionally, recent references should be cited.

English grammar should be improved in this section too such as lines 39-44

- Methods
- Different sections should include more details on how you made such analysis, i.e. software, …etc

- English grammar should be improved in methods this section

Results

The results including figures and tables are well-presented and explained. However, Figure 4 should be improved/replaced with high resolution ones as they can not be read.

Figure 5 also should be improved or replaced with a visible one.

- Discussion
The discussion is somewhat well-written but still needs improvement.
The current data should be compared with previously published findings and how these new findings support the research question.

-References
Up to date references should be included to reveal the up to date information that could support these findings as well.

- Conclusion

Conclusion section should contain the important significant findings revealed in this manuscript as well as the suggested future work recommended to be carried out based on such outcomes and findings.

Author Response

Thank you for your comments. The reply could be found in the attachment.

Reviewer 3 Report

The manuscript by Dilnur et al. reported GWAS analysis of salt related traits using SNP markers for 215 accessions of Asiatic cotton. They focused on nine SNP rich regions distributed 40 candidate genes associated with salt related traits. In general, the manuscript contains information that can benefit to the cotton community. However, I would like to provide some suggestions (major revision) to the authors:

Major comments

1. The author did not show the total SNPs used for GWAS, and why they used the threshold of log10p>=4.0 for manhattan plots? In line 117-118, the author mentioned that most of the upward deviation from the linear line occurred at around –log10p=4.0. This was not the correct reason. There should be a corresponding formula for that.

2. In figure 2, –log10P values and QQ plots suggested relatively weak genetic association for RGA and CIST. The author did not state the reason, and they should use other model to anlysis these data.  

3. The method section should provide the statistical methods (biological replicates and multiple testing) to ensure that this aspect of the manuscript is complete and clear. However, according to the MM part, the author did only one biological replicate for the phenotypic data.

4. The authors detected some relatively significant SNPs; however there is no comparison between the results in the present study and previous studies. It may give an efficient way to state reliable (SNPs) QTLs for salt stress (there were some QTL-mapping studies).

5. The relative values of the results (GR, FW, SL, RWC, ChlC, REC, and MDA) must be consistent throughout the manuscript (In results part, the author mentioned that the GR, FW, SL, WC, ChlC, EC, and MDA were measured. 
However, in MM part. The author mostly stated the RGR, RFW, RSL, RWC, RChlC, REC and RMDA). According to the MM part, the relative value for REC and RWC used in GWAS analysis should be RREC and RRWC.

6. In line 473, CIST = positive index (GR+SL+FW+ChlC+WC)/negative index (EC+MDA). That was not corrected. The author should use 
 CIST = positive index (RGR+RSL+RFW+RChlC+RRWC)/ negative index   (RREC+RMDA) as mentioned above.

7. The values of traits are measured as percentage (bounded between 0 and 1). This type of traits often does not distribute normally and sometime a transformation is needed. Typical transformation for percentage data is y = arcsin (sqrt (x)). Without such a transformation is OK, but the authors should at least mention such a transformation when dealing with % traits. 

8. It is not very necessary however it would be beneficial for breeding strategies to discuss if there is any genotype(s) that are strong under salt stress. 

9. The manuscript included two means of group, one for the accessions (e.g. line 94) and another for candidate region (e.g. line 148). The author should use different name for them.

10. Many references about the methods were needed in the MM part.

11. Although the English is understandable, the manuscript would still benefit from professional language editing

Author Response

Thank you for your comments. The reply could be found in this attachment.

Round 2

Reviewer 2 Report

The authors addressed my concerns

Reviewer 3 Report

Clean version should be attached. It's not very visualized using tracking version.